# A Frank-Wolfe Framework for Efficient and Effective Adversarial Attacks

## Abstract

Depending on how much information an adversary can access to, adversarial attacks can be classified as white-box attack and black-box attack. In both cases, optimization-based attack algorithms can achieve relatively low distortions and high attack success rates. However, they usually suffer from poor time and query complexities, thereby limiting their practical usefulness. In this work, we focus on the problem of developing efficient and effective optimization-based adversarial attack algorithms. In particular, we propose a novel adversarial attack framework for both white-box and black-box settings based on the non-convex Frank-Wolfe algorithm. We show in theory that the proposed attack algorithms are efficient with an $O(1/\sqrt{T})$ convergence rate. The empirical results of attacking Inception V3 model and ResNet V2 model on the ImageNet dataset also verify the efficiency and effectiveness of the proposed algorithms. More specific, our proposed algorithms attain the highest attack success rate in both white-box and black-box attacks among all baselines, and are more time and query efficient than the state-of-the-art.

## 1 Introduction

Deep Neural Networks (DNNs) have made many breakthroughs in different areas of artificial intelligence such as image classification (Krizhevsky et al., 2012; He et al., 2016a), object detection (Ren et al., 2015; Girshick, 2015), and speech recognition (Mohamed et al., 2012; Bahdanau et al., 2016). However, recent studies show that deep neural networks can be vulnerable to adversarial examples (Szegedy et al., 2013; Goodfellow et al., 2015) – a tiny perturbation on an image that is almost invisible to human eyes could mislead a well-trained image classifier towards misclassification. Soon later this is proved to be not a coincidence: similar phenomena have been observed in other problems such as speech recognition (Carlini et al., 2016), visual QA (Xu et al., 2017), image captioning (Chen et al., 2017a), machine translation (Cheng et al., 2018), reinforcement learning (Pattanaik et al., 2018), and even on systems that operate in the physical world (Kurakin et al., 2016).

Depending on how much information an adversary can access to, adversarial attacks can be classified into two classes: white-box attack (Szegedy et al., 2013; Goodfellow et al., 2015) and black-box attack (Papernot et al., 2016a; Chen et al., 2017c). In the white-box setting, the adversary has full access to the target model, while in the black-box setting, the adversary can only access the input and output of the target model but not its internal configurations. Among the approaches proposed for white-box and black-box attacks, optimization-based methods (Carlini & Wagner, 2017; Chen et al., 2017b;c; Ilyas et al., 2018) are most effective: they usually achieve relatively low distortions and high attack success rates. However, these methods are far from efficient. In the white-box setting, they need to solve constrained optimization problems (Carlini & Wagner, 2017), and are usually significantly slower than Fast Gradient Sign Method (FGSM) (Goodfellow et al., 2015) or Iterative FGSM (I-FGM) (Kurakin et al., 2016). Applying those methods with one or two examples are fine, yet in the case of attacking hundreds of thousands examples, e.g. in adversarial training (Kurakin et al., 2016; Madry et al., 2018), this is far from satisfactory.

In the black-box setting, it becomes even more severe since they need to make gradient estimations (Chen et al., 2017c). Therefore, a large number of queries are needed for them to perform a successful attack, especially when the data dimension is large. For example, attacking a $299 \times 299 \times 3$ Imagenet image may take them hundreds of thousands of queries. This significantly limits their prac-

tical usefulness since they can be easily defeated by limiting the number of queries that an adversary can make to the target model.

In this study, we aim to examine the following questions in this study:

*Can we improve the efficiency of the optimization-based attack algorithms? In other words, can we use less time and queries to conduct adversarial attacks?*

In this work, we provide an affirmative answer to this question by proposing an efficient Frank-Wolfe optimization framework for both white-box and black-box attacks. In summary, we make the following main contributions:

- We propose a novel Frank-Wolfe based adversarial attack framework. The white-box attack algorithm is an iterative first-order method which admits the fast gradient sign method (FGSM) as the one-step special case. And the corresponding black-box attack algorithm adopts zeroth-order optimization with two sensing vector options (either from the Euclidean unit sphere or from the standard Gaussian distribution) provided.
- We show that the proposed white-box and black-box attack algorithms enjoy an $O(1/\sqrt{T})$ convergence rate. Also we show that the query complexity of the proposed black-box attack algorithm is linear in data dimension $d$.
- Our empirical results on attacking Inception V3 model with the ImageNet dataset show that (i) the proposed white-box attack algorithm is more efficient than all the baseline white-box algorithms evaluated here, and (ii) the proposed black-box attack algorithm is highly efficient and is also the only one algorithm that achieves a $100\%$ attack success rate.

## 2 RELATED WORK

There is a large body of work on adversarial attacks. In this section, we review the most relevant work in both white-box and black-box attack settings, as well as the non-convex Frank-Wolfe optimization.

**White-box Attacks:** Szegedy et al. (2013) proposed to use box-constrained L-BFGS algorithm for conducting white-box attacks. Goodfellow et al. (2015) proposed the Fast Gradient Sign Method (FGSM) based on linearization of the network as a simple alternative to L-BFGS. Kurakin et al. (2016) proposed to iteratively perform one-step FGSM (Goodfellow et al., 2015) algorithm and clips the adversarial point back to the distortion limit after every iteration. It is called Basic Iterative Method (BIM) or I-FGM in the literature. Madry et al. (2018) showed that for the $L_\infty$ norm case, BIM/I-FGM is equivalent to Projected Gradient Descent (PGD), which is a standard tool for constrained optimization. Papernot et al. (2016b) proposed JSMA to greedily attack the most significant pixel based on the Jacobian-based saliency map. Moosavi-Dezfooli et al. (2016) proposed attack methods by projecting the data to the closest separating hyperplane. Carlini & Wagner (2017) introduced the so-called CW attack by proposing multiple new loss functions for generating adversarial examples. Chen et al. (2017b) followed CW's framework and use an Elastic Net term as the distortion penalty.

**Black-box Attacks:** One popular family of black-box attacks (Hu & Tan, 2017; Papernot et al., 2016a; 2017) is based on the transferability of adversarial examples (Liu et al., 2018; Bhagoji et al., 2017), where an adversarial example generated for one DNN may be reused to attack other neural networks. This allows the adversary to construct a substitute model that mimics the targeted DNN, and then attack the constructed substitute model using white-box attack methods. However, this type of attack algorithms usually suffer from large distortions and relatively low success rates (Chen et al., 2017c). To address this issue, Chen et al. (2017c) proposed the Zeroth-Order Optimization (ZOO) algorithm that extends the CW attack to the black-box setting and uses a zeroth-order optimization approach to conduct the attack. Although ZOO achieves much higher attack success rates than the substitute model-based black-box attacks, it suffers from a poor query complexity since its naive implementation requires to estimate the gradients of all the coordinates (pixels) of the image. To improve its query complexity, several approaches have been proposed. For example, Tu et al. (2018) introduces an adaptive random gradient estimation algorithm and a well-trained Autoencoder to speed up the attack process. Ilyas et al. (2018) and Liu et al. (2018) improved ZOO's query complexity by using Natural Evolutionary Strategies (NES) (Wierstra et al., 2014; Salimans et al., 2017) and active learning, respectively.

**Non-convex Frank-Wolfe Algorithms:** The Frank-Wolfe algorithm (Frank & Wolfe, 1956), also known as the conditional gradient method, is an iterative optimization method for constrained optimization problem. Jaggi (2013) revisited Frank-Wolfe algorithm in 2013 and provided a stronger and more general convergence analysis in the convex setting. Yu et al. (2017) proved the first convergence rate for Frank-Wolfe type algorithm in the non-convex setting. Lacoste-Julien (2016) provided the convergence guarantee for Frank-Wolfe algorithm in the non-convex setting with adaptive step sizes. Reddi et al. (2016) further studied the convergence rate of non-convex stochastic Frank-Wolfe algorithm in the finite-sun optimization setting. Very recently, Staib & Jegelka (2017) proposed to use Frank-Wolfe for distributionally robust training (Sinha et al., 2018). Balasubramanian & Ghadimi (2018) proved the convergence rate for zeroth-order nonconvex Frank-Wolfe algorithm using one-side finite difference gradient estimator with standard Gaussian sensing vectors.

## 3 METHODOLOGY

### 3.1 NOTATIONS

Throughout the paper, scalars are denoted by lower case letters, vectors by lower case bold face letters and sets by calligraphy upper cae letters. For a vector $\mathbf{x} \in \mathbb{R}^d$, we denote the $L_p$ norm of $\mathbf{x}$ by $\|\mathbf{x}\|_p = (\sum_{i=1}^d x_i^p)^{1/p}$. Specially, for $p = \infty$, the $L_\infty$ norm of $\mathbf{x}$ by $\|\mathbf{x}\|_\infty = \max_{i=1}^d |\theta_i|$. We denote $\mathcal{P}_\mathcal{X}(\mathbf{x})$ as the projection operation of projecting vector $\mathbf{x}$ into the set $\mathcal{X}$.

### 3.2 PROBLEM FORMULATION

According to the attack purposes, attacks can be divided into two categories: *untargeted attack* and *targeted attack*. In particular, untargeted attack aims to turn the prediction into any incorrect label, while the targeted attack, which is considerably harder, requires to mislead the classifier to a specific target class. In this work, we follow the literature (Carlini & Wagner, 2017; Ilyas et al., 2018) and focus on the strictly harder targeted attack setting. It is worth noting that our proposed algorithm can be extended to untargeted attack straightforwardly.

Let us define $f(\cdot)$ as the classification loss function of the targeted DNN. For targeted attacks, we aim to learn an adversarial example $\mathbf{x}$ that is close enough to the original input $\mathbf{x}_{\mathrm{ori}}$ and can be misclassified to the target class $y_{\mathrm{tar}}$. The corresponding optimization problem [1] is defined as:

$$\begin{aligned}
\min_\mathbf{x} \quad & f(\mathbf{x}, \, y_{\mathrm{tar}}) \\
\text{subject to} \quad & \|\mathbf{x} - \mathbf{x}_{\mathrm{ori}}\|_p \leq \epsilon.
\end{aligned} \tag{3.1}$$

Evidently, the constraint set $\mathcal{X} := \{\mathbf{x} \mid \|\mathbf{x} - \mathbf{x}_{\mathrm{ori}}\|_p \leq \epsilon\}$ is a bounded convex set when $p \geq 1$. Normally, $p = 2$ and $p = \infty$ are used to measure the distortions $\|\mathbf{x} - \mathbf{x}_{\mathrm{ori}}\|_p$, resulting in $L_2$ attack model and $L_\infty$ attack model respectively. In this work, we study both attack models. In the sequel, since we mainly focus on the targeted attack case, we use $f(\mathbf{x})$ to denote $f(\mathbf{x}, \, y_{\mathrm{tar}})$ for simplicity.

### 3.3 FRANK-WOLFE WHITE-BOX ATTACKS

Frank-Wolfe algorithm (Frank & Wolfe, 1956), also known as the conditional gradient descent, is a popular optimization tool for constrained optimization. Different from PGD that first performs gradient descent followed by a projection step at each iteration, Frank-Wolfe algorithm calls a Linear Minimization Oracle (LMO) over the the constraint set $\mathcal{X}$ at each iteration, i.e.,

$$\mathrm{LMO} \in \operatorname*{argmin}_{\mathbf{v} \in \mathcal{X}} \langle \mathbf{v}, \nabla f(\mathbf{x}_t) \rangle.$$

The LMO can be seen as the minimization of the first-order Taylor expansion of $f(\cdot)$ at point $\mathbf{x}_t$:

$$\min_{\mathbf{v} \in \mathcal{X}} f(\mathbf{x}_t) + \langle \mathbf{v} - \mathbf{x}_t, \nabla f(\mathbf{x}_t) \rangle.$$

By calling LMO, Frank Wolfe solves the linear problem in $\mathcal{X}$ and then perform weighted average with previous iterate to obtain the final update formula.

We present our proposed Frank-Wolfe white-box attack algorithm in Algorithm 1, which is built upon the original Frank-Wolfe algorithm. The key difference between Algorithm 1 and the standard Frank-Wolfe algorithm is in Line 4, where the LMO is called over a slightly relaxed constraint set

---

[1] Note that there is usually an additional constraint on the input variable $\mathbf{x}$, e.g., $\mathbf{x} \in [0, 1]^n$ for normalized image inputs.

$\mathcal{X}_\lambda := \{\mathbf{x} \mid \|\mathbf{x} - \mathbf{x}_{\text{ori}}\|_p \leq \lambda\epsilon\}$ with $\lambda \geq 1$, instead of the original constraint set $\mathcal{X}$. When $\lambda = 1$, set $\mathcal{X}_\lambda$ reduces to $\mathcal{X}$, and Algorithm 1 reduces to standard Frank Wolfe. We argue that this modification makes our algorithm more general, and gives rise to better attack results.

---

**Algorithm 1** Frank-Wolfe White-box Attack Algorithm

---

1: **input:** number of iterations $T$, step sizes $\{\gamma_t\}$, $\lambda > 0$, original image $\mathbf{x}_{\text{ori}}$;
2: $\mathbf{x}_0 = \mathbf{x}_{\text{ori}}$
3: **for** $t = 0, \ldots, T - 1$ **do**
4:     $\mathbf{v}_t = \text{argmin}_{\mathbf{v} \in \mathcal{X}_\lambda} \langle \mathbf{v}, \nabla f(\mathbf{x}_t) \rangle$   // LMO
5:     $\mathbf{d}_t = \mathbf{v}_t - \mathbf{x}_t$
6:     $\mathbf{x}_{t+1} = \mathbf{x}_t + \gamma_t \mathbf{d}_t$
7:     **if** $\lambda > 1$ **then**
8:         $\mathbf{x}_{t+1} = \mathcal{P}_{\mathcal{X}}(\mathbf{x}_{t+1})$
9:     **end if**
10: **end for**
11: **output:** $\mathbf{x}_T$

---

The LMO solution itself can be expensive to obtain in general. Fortunately, applying Frank-Wolfe to solve (3.1) actually gives us a closed-form LMO solution. We provide the solutions of LMO (Line 4 in Algorithm 1) for $L_2$ norm and $L_\infty$ norm cases respectively:

$$\mathbf{v}_t = -\frac{\lambda\epsilon \cdot \nabla f(\mathbf{x}_t)}{\|\nabla f(\mathbf{x}_t)\|_2} + \mathbf{x}_{\text{ori}}, \qquad (L_2 \text{ norm})$$

$$\mathbf{v}_t = -\lambda\epsilon \cdot \text{sign}(\nabla f(\mathbf{x}_t)) + \mathbf{x}_{\text{ori}}. \qquad (L_\infty \text{ norm})$$

The derivation can be found in the supplemental materials.

Note that when $T = 1, \lambda = 1$, substituting the above LMO solutions into Algorithm 1 yields the final update of $x_1 = x_0 - \gamma_t \epsilon \cdot \nabla f(\mathbf{x}_t)$, which reduces to FGSM [2] when $\gamma_t = 1$. Similar derivation also applies to $L_2$ norm case. Therefore, just like PGD, our proposed Frank-Wolfe white-box attack also includes FGSM (FGM) as a one-step special instance.

### 3.4 FRANK-WOLFE BLACK-BOX ATTACKS

Next we consider the black-box setting, where we cannot perform back-propagation to calculate the gradient of the loss function anymore. Instead, we can only query the DNN system's outputs with specific inputs. To clarify, here the output refers to the logit layer's output (confidence scores for classification), not the final prediction label. The label-only setting is doable under our framework, but will incur extra difficulty such as designing new loss functions. For simplicity, here we consider the confidence score output.

We propose a zeroth-order Frank-Wolfe based algorithm to solve this problem. Algorithm 2 show our proposed Frank-Wolfe black-box attack algorithm. The key difference between our proposed black-box attack and white-box attack is one extra gradient estimation step, which is presented in Line 4 in Algorithm 2. Also note that for the final output, we provide two options. While option II is the common choice in practice, option I is also provided for the ease of theoretical analysis.

As many other zeroth-order optimization algorithms (Shamir, 2017; Flaxman et al., 2005), Algorithm 3 uses symmetric finite differences to estimate the gradient and therefore, gets rid of the dependence on back-propagation in white-box setting. Different from Chen et al. (2017c), here we do not utilize natural basis as our sensing vectors, instead, we provide two options: one is to use vectors uniformly sampled from Euclidean unit sphere and the other is to use vectors uniformly sampled from standard multivarite Gaussian distribution. This will greatly improve the gradient estimation efficiency comparing to sensing with natural basis as such option will only be able to estimate one coordinate of the gradient vector per query. In practice, both options here provide us competitive experimental results. It is worth noting that NES method (Wierstra et al., 2014) with antithetic sampling (Salimans et al., 2017) used in Ilyas et al. (2018) yields similar formula as our Option II in Algorithm 3.

---

[2]The extra clipping operation in FGSM is to project to the additional box constraint for image classification task. We will also need this clipping operation at the end of each iteration for specific tasks such as image classification.

---

**Algorithm 2** Frank-Wolfe Black-box Attack Algorithm

---

1: **input:** number of iterations $T$, step sizes $\{\gamma_t\}$, $\lambda > 0$, original image $\mathbf{x}_{\mathrm{ori}}$, target label $y_{\mathrm{tar}}$;
2: $\mathbf{x}_0 = \mathbf{x}_{\mathrm{ori}}$
3: **for** $t = 0, \ldots, T - 1$ **do**
4: $\quad$ $\mathbf{q}_t = \mathrm{ZERO\_ORD\_GRAD\_EST}(\mathbf{x}_t)$ // Algorithm 3
5: $\quad$ $\mathbf{v}_t = \mathrm{argmin}_{\mathbf{v} \in \mathcal{X}_\lambda} \langle \mathbf{v}, \mathbf{q}_t \rangle$
6: $\quad$ $\mathbf{d}_t = \mathbf{v}_t - \mathbf{x}_t$
7: $\quad$ $\mathbf{x}_{t+1} = \mathbf{x}_t + \gamma_t \mathbf{d}_t$
8: $\quad$ **if** $\lambda > 1$ **then**
9: $\quad\quad$ $\mathbf{x}_{t+1} = \mathcal{P}_{\mathcal{X}}(\mathbf{x}_{t+1})$
10: $\quad$ **end if**
11: **end for**
12: **Option I:** $\mathbf{x}_a$ is uniformly random chosen from $\{\mathbf{x}_t\}_{t=1}^T$
13: **Option II:** $\mathbf{x}_a = \mathbf{x}_T$
14: **output:** $\mathbf{x}_a$

---

**Algorithm 3** Zeroth-Order Gradient Estimation (ZERO\_ORD\_GRAD\_EST)

---

1: **parameters:** number of gradient estimation samples $b$, sampling parameter $\delta_t$;
2: $\mathbf{q} = \mathbf{0}$
3: **for** $i = 1, \ldots, b$ **do**
4: $\quad$ **Option I:** Sample $\mathbf{u}_i$ uniformly from the Euclidean unit sphere with $\|\mathbf{u}_i\|_2 = 1$
$$\mathbf{q} = \mathbf{q} + \frac{d}{2\delta_t b}\big(f(\mathbf{x}_t + \delta_t \mathbf{u}_i) - f(\mathbf{x}_t - \delta_t \mathbf{u}_i)\big)\mathbf{u}_i$$
5: $\quad$ **Option II:** Sample $\mathbf{u}_i$ uniformly from the standard Gaussian distribution $\mathcal{N}(\mathbf{0}, \mathbf{I})$
$$\mathbf{q} = \mathbf{q} + \frac{1}{2\delta_t b}\big(f(\mathbf{x}_t + \delta_t \mathbf{u}_i) - f(\mathbf{x}_t - \delta_t \mathbf{u}_i)\big)\mathbf{u}_i$$
6: **end for**
7: **output:** $\mathbf{q}$

---

## 4 MAIN THEORY

In this section, we establish the convergence guarantees for our proposed Frank-Wolfe adversarial attack algorithms described in Section 3. First, we introduce the convergence criterion for our Frank-Wolfe adversarial attack framework.

### 4.1 CONVERGENCE CRITERION

The loss function for common DNN models are generally nonconvex. In addition, (3.1) is a constrained optimization. For such general nonconvex constrained optimization, we typically adopt the Frank-Wolfe gap as the convergence criterion (since gradient norm of $f$ is no longer a proper criterion for constrained optimization problems):

$$g(\mathbf{x}_t) = \max_{\mathbf{x} \in \mathcal{X}} \langle \mathbf{x} - \mathbf{x}_t, -\nabla f(\mathbf{x}_t) \rangle.$$

Note that for the Frank-Wolfe gap, we always have $g(\mathbf{x}_t) \geq 0$ and $\mathbf{x}_t$ is a stationary point for the constrained optimization problem if and only if $g(\mathbf{x}_t) = 0$. Also the Frank-Wolfe gap is affine invariant and do not tie to any specific choice of norm, which makes itself a perfect convergence criterion for Frank-Wolfe based algorithms.

### 4.2 CONVERGENCE GUARANTEE FOR FRANK-WOLFE WHITE-BOX ATTACK

Before we are going to provide the convergence guarantee of Frank-Wolfe white-box attack (Algorithm 1), we introduce the following assumptions that are essential to the convergence analysis.

**Assumption 4.1.** Function $f(\cdot)$ is $L$-smooth with respect to $\mathbf{x}$, i.e., for any $\mathbf{x}, \mathbf{x}'$, it holds that

$$f(\mathbf{x}') \leq f(\mathbf{x}) + \nabla f(\mathbf{x})^\top (\mathbf{x}' - \mathbf{x}) + \frac{L}{2}\|\mathbf{x}' - \mathbf{x}\|_2^2.$$

Assumption 4.1 is a standard assumption in nonconvex optimization, and is also adopted in other Frank-Wolfe literature such as Lacoste-Julien (2016); Reddi et al. (2016). Note that even though the smoothness assumption does not hold for general DNN models, a recent study (Santurkar et al., 2018) shows that batch normalization that is used in many modern DNNs such as Inception V3

model, actually makes the optimization landscape significantly smoother [3]. This justifies the validity of Assumption 4.1.

**Assumption 4.2.** Set $\mathcal{X}$ is bounded with diameter $D$, i.e., $\|\mathbf{x} - \mathbf{x}'\|_2 \leq D$ for all $\mathbf{x}, \mathbf{x}' \in \mathcal{X}$.

Assumption 4.2 implies that the input space is bounded. For common tasks such as image classification, given the fact that images have bounded pixel range and $\epsilon$ is a small constant, this assumption trivially holds.

Now we present the theorem, which characterizes the convergence rate of our proposed Frank-Wolfe white-box adversarial attack algorithm presented in Algorithm 1.

**Theorem 4.3.** Under Assumptions 4.1 and 4.2, let $\gamma_t = \gamma = \sqrt{2(f(\mathbf{x}_0) - f(\mathbf{x}^*))/(LD^2 T)}$, denote $\widetilde{g}_T = \min_{1 \leq k \leq T} g(\mathbf{x}_k)$ where $\{\mathbf{x}_k\}_{k=1}^T$ are iterates in Algorithm 1 with $\lambda = 1$, we have:

$$\widetilde{g}_T \leq \sqrt{\frac{LD^2(f(\mathbf{x}_0) - f(\mathbf{x}^*))}{2T}},$$

where $\mathbf{x}^*$ is the optimal solution to (3.1).

**Remark 4.4.** Theorem 4.3 suggests that our proposed Frank-Wolfe white-box attack algorithm achieves a $O(1/\sqrt{T})$ rate of convergence. Note that similar result has been proved in Lacoste-Julien (2016) under a different choice of step size.

### 4.3 CONVERGENCE GUARANTEE FOR FRANK-WOLFE BLACK-BOX ATTACK

Next we analyze the convergence of our proposed Frank-Wolfe black-box adversarial attack algorithm presented in Algorithm 2.

In order to prove the convergence of our proposed Frank-Wolfe black-box attack algorithm, we need the following additional assumption that $\|\nabla f(\mathbf{0})\|_2$ is bounded.

**Assumption 4.5.** Gradient of $f(\cdot)$ at zero point $\nabla f(\mathbf{0})$ satisfies $\max_y \|\nabla f(\mathbf{0})\|_2 \leq C_g$.

Following the analysis in Shamir (2017), let $f_\delta(\mathbf{x}) = \mathbb{E}_{\mathbf{u}}[f(\mathbf{x} + \delta\mathbf{u})]$, which is the smoothed version of $f(\mathbf{x})$. This smoothed function value plays a central role in our theoretical analysis, since it bridges the finite difference gradient approximation with the actual gradient. The following lemma shows this relationship.

**Lemma 4.6.** For the gradient estimator $\mathbf{q}_t$ in Algorithm 3, its expectation and variance satisfy

$$\mathbb{E}[\mathbf{q}_t] = \nabla f_\delta(\mathbf{x}_t), \qquad \mathbb{E}\|\mathbf{q}_t - \mathbb{E}[\mathbf{q}_t]\|_2^2 \leq \frac{1}{b}\left(2d(C_g + LD)^2 + \frac{1}{2}\delta_t^2 L^2 d^2\right).$$

Now we are going to present the theorem, which characterizes the convergence rate of Algorithm 2.

**Theorem 4.7.** Under Assumptions 4.1, 4.2 and 4.5, let $\gamma_t = \gamma = \sqrt{2(f(\mathbf{x}_0) - f(\mathbf{x}^*))/(LD^2 T)}$, $b = Td$ and $\delta_t = \sqrt{2/Td^2}$, suppose we use Option I in Algorithm 2 and option II for Algorithm 3, then the output $\mathbf{x}_a$ from Algorithm 2 with $\lambda = 1$ satisfies:

$$\mathbb{E}[g(\mathbf{x}_a)] \leq \frac{D}{\sqrt{2T}}\left(\sqrt{L(f(\mathbf{x}_0) - f(\mathbf{x}^*))} + 2(L + C_g + LD)\right),$$

where $\mathbf{x}^*$ is the optimal solution to (3.1).

**Remark 4.8.** Theorem 4.7 suggests that Algorithm 2 also enjoys a $O(1/\sqrt{T})$ rate of convergence. In terms of query complexity, the total number of queries needed is $Tb = T^2 d$, which is linear in the data dimension $d$. In fact, in the experiment part, we observed that this number can be substantially smaller than $d$, e.g., $b = 25$, which is much lower than the theorem suggests. Note that although we only prove for option I in Algorithm 3, our result can be readily extended to Option II (the Gaussian sensing vector case).

---

[3]The original argument in Santurkar et al. (2018) refers to the smoothness with respect to each layer's parameters. Note that the first layer's parameters are in the mirror position (in terms of backpropagation) as the network inputs. Therefore, the argument in Santurkar et al. (2018) can also be applied here with respect to the network inputs.

## 5 EXPERIMENTS

In this section, we present the experimental results for our proposed Frank-Wolfe attack framework against other state-of-the-art adversarial attack algorithms in both white-box and black-box settings. All of our experiments are conducted on Amazon AWS p3.2xlarge servers which come with Intel Xeon E5 CPU and one NVIDIA Tesla V100 GPU (16G RAM). All experiments are implemented in Tensorflow platform version 1.10.0 within Python 3.6.4.

### 5.1 EVALUATION SETUP AND METRICS

We test the attack effectiveness of all algorithms by evaluating on a pre-trained Inception V3 model (Szegedy et al., 2016) and a ResNet V2 50 (He et al., 2016b) model that are trained on ImageNet dataset (Deng et al., 2009). The pre-trained Inception V3 model is reported to have a $78.0\%$ top-1 accuracy and a $93.9\%$ top-5 accuracy. The pre-trained ResNet V2 model is reported to have a $75.6\%$ top-1 and a $92.8\%$ top-5 accuracy. We randomly choose 500 images from the ImageNet validation set that are verified to be correctly classified by the pre-trained model and also randomly choose a target class for each image. Each image has a dimension of $299 \times 299 \times 3$ and we test all attack algorithms through the same randomly chosen data samples and target labels.

We test for both $L_2$ norm based and $L_\infty$ norm based attacks. In the white-box setting, we perform binary search / grid search for the best distortion parameter ($\epsilon$ in our formulation and $c$ in CW's regularized formulation). In the black-box setting, for $L_2$ norm based attack, we set $\epsilon = 5$ and for $L_\infty$ based attack, we set $\epsilon = 0.05$. For white-box attack, we restrict a maximum of $1,000$ iterations per attack for each method. And for black-box attack, we set a maximum query limit of $500,000$ per attack per image for each method.

For all algorithms, we stop the algorithm when a successful attack is found. For our proposed black-box attack, we use option II in Algorithm 2 and test both options in Algorithm 3. We set the number of gradient estimation samples $b = 25$ for Algorithm 2. More detailed description on parameter settings can be found in the supplemental materials.

We evaluate the final performance through attack success rate where the success is defined as making the classifier output the exact target class label (not any incorrect labels). We also measure average attack time per image, average distortion (only on successful attacked samples) and average number of queries needed (only for black-box attack) per image. For a fair time comparison, even though some of the algorithms including ours can be written in batch form (attack multiple images at one time), all algorithms are set to attack one image at a time.

Due to page limit, we leave all experimental results on ResNet V2 model in the supplemental materials.

### 5.2 BASELINE METHODS

We compare the proposed algorithms with several state-of-the-art baseline algorithms. Specifically, we compare the proposed white-box attack algorithm with [4] (i) PGD (Madry et al., 2018) (which is essentially I-FGM (Kurakin et al., 2016)), (ii) CW attack (Carlini & Wagner, 2017) and (iii) EAD attack (Chen et al., 2017b). We compare the proposed black-box attack algorithm with (i) ZOO attack (Chen et al., 2017c) and (ii) NES-PGD attack (Ilyas et al., 2018).

### 5.3 WHITE-BOX ATTACK EXPERIMENTS

In this subsection, we present the white-box attack experiments on Inception V3 model. Tables 1 and 2 present our experimental results for $L_2$ norm and $L_\infty$ norm based white-box attacks respectively. As we can observe from the tables, the attack success rate is $100\%$ for every method. For the other baselines in the $L_2$ norm case, CW method achieves the smallest average distortion, yet it comes with an expansive time cost. EAD method does not have either time advantage or distortion advantage in this experiment, probably due to its different motivation in attacking. PGD has moderate average distortion, yet it also costs quite some time to finish the attack. On the other hand, our proposed algorithm achieves the shortest attack time with moderate distortion. It significantly reduces the time complexity needed for attacking data with large dimensionality. For the $L_\infty$ norm case, CW method takes significantly longer time and does not perform very well on average distortion either.

---

[4]We did not compare with FGM (FGSM) (Goodfellow et al., 2015) since it basically has zero success rate for targeted attack on Inception V3 or ResNet V2 models.

Table 1: Comparison of $L_2$ norm based white-box attacks on Inception V3 model with $\epsilon = 5$. We report attack success rate, average time and average distortion.

| METHODS | SUCCESS RATE (%) | AVERAGE TIME (s) | AVERAGE DISTORTION |
|---|---|---|---|
| PGD | 100.0 | 143.2 | 0.74 |
| CW | 100.0 | 169.9 | **0.57** |
| EAD | 100.0 | 167.8 | 1.09 |
| FW-White | 100.0 | **50.6** | 0.85 |

Table 2: Comparison of $L_\infty$ norm based white-box attacks on Inception V3 model with $\epsilon = 0.05$. We report attack success rate, average time and average distortion.

| METHODS | SUCCESS RATE (%) | AVERAGE TIME (s) | AVERAGE DISTORTION |
|---|---|---|---|
| PGD | 100.0 | 39.1 | **0.0027** |
| CW | 100.0 | 745.2 | 0.0071 |
| FW-White | 100.0 | **13.7** | 0.0034 |

This is largely due to the original CW was designed for $L_2$ norm attack, and in order to apply it to $L_\infty$ norm attack, special design is needed, which sacrifices its performance in terms of runtime. Again, our proposed white-box attack algorithm achieves the shortest average attack time and a moderate average distortion.

In Figure 1, we also examine the effect of $\lambda$ in our proposed Frank-Wolfe white-box attack algorithm. We plot the objective loss function value of attacking one example against the number of iterations for both $L_2$ and $L_\infty$ based white-box attack on Inception V3 model. From the plot, we can see that larger $\lambda$ indeed leads to faster convergence.

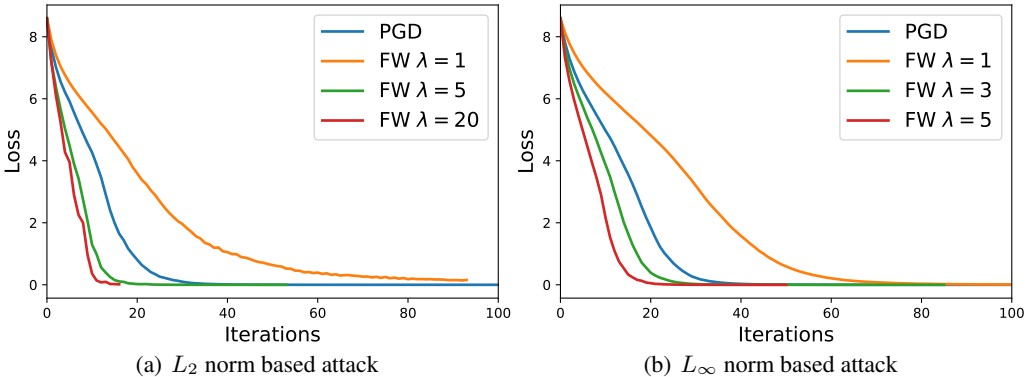

(a) $L_2$ norm based attack ‎ ‎ ‎ ‎ ‎ ‎ ‎ ‎ ‎ ‎ (b) $L_\infty$ norm based attack

Figure 1: Loss against the number of iterations plot for PGD and FW algorithms in both $L_2$ norm and $L_\infty$ norm based white-box attacks on Inception V3 model.

### 5.4 BLACK-BOX ATTACK EXPERIMENTS

In this subsection, we present the black-box attack experiments on Inception V3 model. For black-box attacks, attack success rate, time and number of queries needed are more meaningful evaluation metrics than distortion distances. Therefore, we omit all the grid search / binary search steps that are used in the white-box setting since extra time / queries are needed for finding parameters that can obtain better distortion distances.

Tables 3 and 4 present our experimental results for $L_2$ norm and $L_\infty$ norm based black-box attacks respectively. For ZOO method, note that it only has the $L_2$ norm version and it follows CW's framework and thus uses different loss function and problem formulation (cannot exactly control the adversarial example to be within the distortion limit, we manage to keep the average distortion around $\epsilon$ for ZOO while other methods have average distortions very close to $\epsilon$). Furthermore, we can observe that ZOO is quite slow in this task. Attack on a single image can take up to 2 hours for ZOO and it is only able to achieve a 74.8% success rate (compared with the 88.9% success rate in the original paper, we think the main reason is the query limit here is only half of the query limit

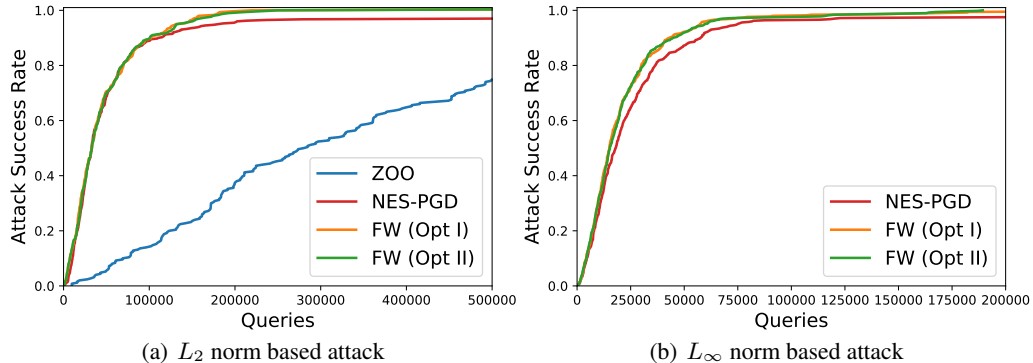

(a) $L_2$ norm based attack                    (b) $L_\infty$ norm based attack

Figure 2: Attack success rate against the number of queries plot for different black-box attack algorithms in both $L_2$ norm $L_\infty$ norm cases on Inception V3 model.

in the original paper). NES-PGD method, while greatly improving ZOO's performance, still cannot achieve $100\%$ success rate in both attack models and takes relatively more time and queries. In sharp contrast, our proposed Frank-Wolfe black-box attacks (both option I and option II) achieve the highest success rate in both $L_2$ norm and $L_\infty$ norm based black-box attacks and further largely improve the attack efficiency.

Table 3: Comparison of $L_2$ norm based black-box attacks on Inception V3 model with $\epsilon = 5$. We report attack success rate, average time and average number of queries needed per image. Opt I and Opt II refer to the two options in Algorithm 2.

| METHODS | SUCCESS RATE (%) | AVERAGE TIME (s) | AVERAGE QUERIES |
|---|---|---|---|
| ZOO | 74.8 | 5692.6 | 296867.0 |
| NES-PGD | 96.7 | 133.0 | 58921.8 |
| FW-Black (Opt I) | **100.0** | 102.9 | 45994.5 |
| FW-Black (Opt II) | **100.0** | **100.9** | **45156.0** |

Table 4: Comparison of $L_\infty$ norm based black-box attacks on Inception V3 model with $\epsilon = 0.05$. We report attack success rate, average time and average number of queries needed per image. Opt I and Opt II refer to the two options in Algorithm 2.

| METHODS | SUCCESS RATE (%) | AVERAGE TIME (s) | AVERAGE QUERIES |
|---|---|---|---|
| NES-PGD | 98.0 | 76.9 | 34062.2 |
| FW-Black (Opt I) | **100.0** | **50.4** | **22313.2** |
| FW-Black (Opt II) | **100.0** | 50.6 | 22424.1 |

Figure 2 illustrates the attack success rate against the number of queries plot for different algorithms in both $L_2$ norm and $L_\infty$ norm based black-box attacks on Inception V3 model. As we can see from the plot, our proposed Frank-Wolfe black-box attack algorithm (both options) achieves the highest attack success rate and best efficiency (least queries needed for achieving the same success rate), especially in the $L_2$ norm case.

## 6  CONCLUSIONS

In this work, we propose a Frank-Wolfe framework for efficient and effective adversarial attacks. Our proposed white-box and black-box attack algorithms enjoy an $O(1/\sqrt{T})$ rate of convergence, and the query complexity of the proposed black-box attack algorithm is linear in data dimension $d$. Finally, our empirical study on attacking Inception V3 model with ImageNet dataset yields a $100\%$ attack success rate for our proposed algorithms, even in the setting of black-box attack.

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

## A  LINEAR MINIMIZATION ORACLE (LMO) SOLUTIONS

Denote $\mathbf{u} = (\mathbf{v} - \mathbf{x}_{\mathrm{ori}})/(\lambda\epsilon)$, the linear minimization problem can be written as

$$
\begin{aligned}
\min_{\|\mathbf{v}-\mathbf{x}_{\mathrm{ori}}\|_p \leq \lambda\epsilon} \langle \mathbf{v}, \nabla f(\mathbf{x}_t) \rangle &= \min_{\|\mathbf{u}\|_p \leq 1} \lambda\epsilon \cdot \langle \mathbf{u}, \nabla f(\mathbf{x}_t) \rangle \\
&= \max_{\|\mathbf{u}\|_p \leq 1} \lambda\epsilon \cdot \langle \mathbf{u}, -\nabla f(\mathbf{x}_t) \rangle \\
&= \lambda\epsilon \cdot \|\nabla f(\mathbf{x}_t)\|_{p*},
\end{aligned}
$$

where $\|\cdot\|_{p*}$ denotes the dual norm of $\|\cdot\|_p$. For $p = 2$ case, we have

$$
\langle (\mathbf{v} - \mathbf{x}_{\mathrm{ori}})/(\lambda\epsilon), -\nabla f(\mathbf{x}_t) \rangle = \|\nabla f(\mathbf{x}_t)\|_2.
$$

It immediately implies that

$$
\mathbf{v} = -\frac{\lambda\epsilon \cdot \nabla f(\mathbf{x}_t)}{\|\nabla f(\mathbf{x}_t)\|_2} + \mathbf{x}_{\mathrm{ori}}.
$$

For $p = \infty$ case, we have

$$
\langle (\mathbf{v} - \mathbf{x}_{\mathrm{ori}})/(\lambda\epsilon), -\nabla f(\mathbf{x}_t) \rangle = \|\nabla f(\mathbf{x}_t)\|_1.
$$

It immediately implies that

$$
\mathbf{v} = -\lambda\epsilon \cdot \mathrm{sign}(\nabla f(\mathbf{x}_t)) + \mathbf{x}_{\mathrm{ori}}.
$$

For the ease of comparison, we show the full update formula (before final projection step) for our algorithm. In detail, for $p = \infty$ case, our algorithm takes the following update formulate:

$$
\begin{aligned}
\mathbf{x}_{t+1} &= (1 - \gamma_t)\mathbf{x}_t + \gamma_t \mathbf{v}_t \\
&= (1 - \gamma_t)\mathbf{x}_t - \lambda\gamma_t\epsilon \cdot \mathrm{sign}(\nabla f(\mathbf{x}_t)) + \gamma_t \cdot \mathbf{x}_{\mathrm{ori}} \\
&= \mathbf{x}_t - \lambda\gamma_t\epsilon \cdot \mathrm{sign}(\nabla f(\mathbf{x}_t)) - \gamma_t(\mathbf{x}_t - \mathbf{x}_{\mathrm{ori}}),
\end{aligned}
$$

and for $p = 2$ case, it takes

$$
\mathbf{x}_{t+1} = \mathbf{x}_t - \lambda\gamma_t\epsilon \cdot \frac{\nabla f(\mathbf{x}_t)}{\|\nabla f(\mathbf{x}_t)\|_2} - \gamma_t(\mathbf{x}_t - \mathbf{x}_{\mathrm{ori}}).
$$

Compared with PGD, the full update (before final projection step) of Frank-Wolfe white-box attack includes an extra parameter $\lambda$ before the normalized gradient, as well as an extra term $(\mathbf{x}_t - \mathbf{x}_{\mathrm{ori}})$. This difference makes the behavior of Frank-Wolfe based attacks different from that of PGD based attacks.

## B  PROOF OF THE MAIN THEORY IN SECTION 4

### B.1  PROOF OF THEOREM 4.3

*Proof.* For simplicity, we denote $f(\mathbf{x}_t)$ by $f(\mathbf{x}_t)$ for the rest of the proof. First by Assumption 4.1, we have

$$
\begin{aligned}
f(\mathbf{x}_{t+1}) &\leq f(\mathbf{x}_t) + \nabla f(\mathbf{x}_t)^\top (\mathbf{x}_{t+1} - \mathbf{x}_t) + \frac{L}{2}\|\mathbf{x}_{t+1} - \mathbf{x}_t\|_2^2 \\
&= f(\mathbf{x}_t) + \gamma \nabla f(\mathbf{x}_t)^\top (\mathbf{v}_t - \mathbf{x}_t) + \frac{L\gamma^2}{2}\|\mathbf{v}_t - \mathbf{x}_t\|_2^2 \\
&\leq f(\mathbf{x}_t) + \gamma \nabla f(\mathbf{x}_t)^\top (\mathbf{v}_t - \mathbf{x}_t) + \frac{LD^2\gamma^2}{2},
\end{aligned}
$$

where the last inequality uses the bounded domain condition in Assumption 4.2. Note that by definition of the Frank-Wolfe gap, we have

$$
f(\mathbf{x}_{t+1}) \leq f(\mathbf{x}_t) - \gamma g(\mathbf{x}_t) + \frac{LD^2\gamma^2}{2}.
$$

Summation over $t$ of the above inequality, we obtain

$$f(\mathbf{x}_T) \le f(\mathbf{x}_0) - \sum_{k=0}^{T-1} \gamma g(\mathbf{x}_k) + \frac{TLD^2\gamma^2}{2}$$

$$\le f(\mathbf{x}_0) - \gamma T \widetilde{g}_T + \frac{TLD^2\gamma^2}{2},$$

where the second inequality follows from the definition of $\widetilde{g}_t$. Note that by optimality we easily have $f(\mathbf{x}_{t+1}) \ge f(\mathbf{x}^*)$. Rearrange the above inequality we have

$$\widetilde{g}_T \le \frac{f(\mathbf{x}_0) - f(\mathbf{x}^*)}{T\gamma} + \frac{LD^2\gamma}{2}$$

$$\le \sqrt{\frac{LD^2(f(\mathbf{x}_0) - f(\mathbf{x}^*))}{2T}},$$

where the second inequality is achieved when $\gamma = \sqrt{2(f(\mathbf{x}_0) - f(\mathbf{x}^*))/(LD^2T)}$. $\qquad\square$

### B.2 PROOF OF LEMMA 4.6

*Proof.* For simplicity we denote $f(\cdot)$ by $f(\cdot)$ for the rest of the proof. Let us denote $\boldsymbol{\psi}_i = \frac{d}{2\delta_t b}\big(f(\mathbf{x}_t + \delta_t \mathbf{u}_i) - f(\mathbf{x}_t - \delta_t \mathbf{u}_i)\big)\mathbf{u}_i$. For the first part, we have

$$\mathbb{E}[\boldsymbol{\psi}_i] = \mathbb{E}_{\mathbf{u}}\left[\frac{d}{2\delta_t b}\big(f(\mathbf{x}_t + \delta_t \mathbf{u}_i) - f(\mathbf{x}_t - \delta_t \mathbf{u}_i)\big)\mathbf{u}_i\right]$$

$$= \mathbb{E}_{\mathbf{u}}\left[\frac{d}{2\delta_t b} f(\mathbf{x}_t + \delta_t \mathbf{u}_i)\mathbf{u}_i\right] + \mathbb{E}_{\mathbf{u}}\left[\frac{d}{2\delta_t b} f(\mathbf{x}_t - \delta_t \mathbf{u}_i)(-\mathbf{u}_i)\right]$$

$$= \mathbb{E}_{\mathbf{u}}\left[\frac{d}{\delta_t b} f(\mathbf{x}_t + \delta_t \mathbf{u}_i)\mathbf{u}_i\right]$$

$$= \frac{1}{b}\nabla f_\delta(\mathbf{x}_t),$$

where the third equality holds due to symmetric property of $\mathbf{u}_i$ and the last equality follows from Lemma 4.1(a) in Gao et al. (2018). Therefore, we have

$$\mathbb{E}[\mathbf{q}_t] = \mathbb{E}\left[\sum_{i=1}^{b} \boldsymbol{\psi}_i\right] = \nabla f_\delta(\mathbf{x}_t).$$

For second part, note that $\boldsymbol{\psi}_i$'s are independent from each other due to the independence of $\mathbf{u}_i$, we have

$$\mathbb{E}\|\mathbf{q}_t - \mathbb{E}[\mathbf{q}_t]\|_2^2 = \mathbb{E}\left\|\sum_{i=1}^{b}\big[\boldsymbol{\psi}_i - \mathbb{E}\boldsymbol{\psi}_i\big]\right\|_2^2 = \sum_{i=1}^{b}\mathbb{E}\big\|\boldsymbol{\psi}_i - \mathbb{E}\boldsymbol{\psi}_i\big\|^2 \le \sum_{i=1}^{b}\mathbb{E}\big\|\boldsymbol{\psi}_i\big\|^2.$$

Now take a look at $\mathbb{E}\big\|\boldsymbol{\psi}_i\big\|^2$:

$$\mathbb{E}\big\|\boldsymbol{\psi}_i\big\|^2 = \mathbb{E}_{\mathbf{u}}\left\|\frac{d}{2\delta_t b}\big(f(\mathbf{x}_t + \delta_t \mathbf{u}_i) - f(\mathbf{x}_t) + f(\mathbf{x}_t) - f(\mathbf{x}_t - \delta_t \mathbf{u}_i)\big)\mathbf{u}_i\right\|_2^2$$

$$\le \frac{1}{2b^2}\mathbb{E}_{\mathbf{u}}\left\|\frac{d}{\delta_t}\big(f(\mathbf{x}_t + \delta_t \mathbf{u}_i) - f(\mathbf{x}_t)\big)\mathbf{u}_i\right\|_2^2 + \frac{1}{2b^2}\mathbb{E}_{\mathbf{u}}\left\|\frac{d}{\delta_t}\big(f(\mathbf{x}_t) - f(\mathbf{x}_t - \delta_t \mathbf{u}_i)\big)\mathbf{u}_i\right\|_2^2$$

$$= \frac{1}{b^2}\mathbb{E}_{\mathbf{u}}\left\|\frac{d}{\delta_t}\big(f(\mathbf{x}_t + \delta_t \mathbf{u}_i) - f(\mathbf{x}_t)\big)\mathbf{u}_i\right\|_2^2$$

$$\le \frac{1}{b^2}\left(2d\|\nabla f(\mathbf{x}_t)\|_2^2 + \frac{1}{2}\delta_t^2 L^2 d^2\right),$$

where the first inequality is due to the fact that $(a+b)^2 \le 2a^2 + 2b^2$, the second equality follows from the symmetric property of $\mathbf{u}_i$ and the last inequality is by Lemma 4.1(b) in Gao et al. (2018). Also note that by Assumption 4.1 and 4.5 we have

$$\|\nabla f(\mathbf{x}_t)\|_2^2 \le (\|\nabla f(\mathbf{0}))\|_2 + L\|\mathbf{x}_t\|_2)^2 \le (C_g + LD)^2.$$

Combine all above results, we obtain

$$\mathbb{E}\|\mathbf{q}_t - \mathbb{E}[\mathbf{q}_t]\|_2^2 \le \frac{1}{b}\left(2d(C_g + LD)^2 + \frac{1}{2}\delta_t^2 L^2 d^2\right).$$

$\square$

### B.3 Proof of Theorem 4.7

*Proof.* For simplicity we denote $f(\mathbf{x}_t)$ by $f(\mathbf{x}_t)$ for the rest of the proof. First by Assumption 4.1, we have

$$f(\mathbf{x}_{t+1}) \le f(\mathbf{x}_t) + \nabla f(\mathbf{x}_t)^\top (\mathbf{x}_{t+1} - \mathbf{x}_t) + \frac{L}{2}\|\mathbf{x}_{t+1} - \mathbf{x}_t\|_2^2$$

$$= f(\mathbf{x}_t) + \gamma \nabla f(\mathbf{x}_t)^\top (\mathbf{v}_t - \mathbf{x}_t) + \frac{L\gamma^2}{2}\|\mathbf{v}_t - \mathbf{x}_t\|_2^2$$

$$\le f(\mathbf{x}_t) + \gamma \nabla f(\mathbf{x}_t)^\top (\mathbf{v}_t - \mathbf{x}_t) + \frac{LD^2\gamma^2}{2}$$

$$= f(\mathbf{x}_t) + \gamma \mathbf{q}_t^\top (\mathbf{v}_t - \mathbf{x}_t) + \gamma(\nabla f(\mathbf{x}_t) - \mathbf{q}_t)^\top (\mathbf{v}_t - \mathbf{x}_t) + \frac{LD^2\gamma^2}{2},$$

where the second inequality uses the bounded domain condition in Assumption 4.2. Now define an auxiliary quantity:

$$\widehat{\mathbf{v}}_t = \operatorname*{argmin}_{\mathbf{v} \in \mathcal{X}} \langle \mathbf{v}, \nabla f(\mathbf{x}_t)\rangle.$$

According to the definition of $g(\mathbf{x}_t)$, this immediately implies

$$g(\mathbf{x}_t) = \langle \widehat{\mathbf{v}}_t, \nabla f(\mathbf{x}_t)\rangle.$$

Then we further have

$$f(\mathbf{x}_{t+1}) \le f(\mathbf{x}_t) + \gamma \mathbf{q}_t^\top (\widehat{\mathbf{v}}_t - \mathbf{x}_t) + \gamma(\nabla f(\mathbf{x}_t) - \mathbf{q}_t)^\top (\mathbf{v}_t - \mathbf{x}_t) + \frac{LD^2\gamma^2}{2}$$

$$= f(\mathbf{x}_t) + \gamma \nabla f(\mathbf{x}_t)^\top (\widehat{\mathbf{v}}_t - \mathbf{x}_t) + \gamma(\nabla f(\mathbf{x}_t) - \mathbf{q}_t)^\top (\mathbf{v}_t - \widehat{\mathbf{v}}_t) + \frac{LD^2\gamma^2}{2}$$

$$= f(\mathbf{x}_t) - \gamma g(\mathbf{x}_t) + \gamma(\nabla f(\mathbf{x}_t) - \mathbf{q}_t)^\top (\mathbf{v}_t - \widehat{\mathbf{v}}_t) + \frac{LD^2\gamma^2}{2}$$

$$\le f(\mathbf{x}_t) - \gamma g(\mathbf{x}_t) + \gamma D \cdot \|\nabla f(\mathbf{x}_t) - \mathbf{q}_t\|_2 + \frac{LD^2\gamma^2}{2},$$

where the first inequality follows from the optimally of $\mathbf{v}_t$ in Algorithm 2 and the last inequality holds due to Cauchy-Schwarz inequality. Take expectations for both sides of the above inequality, we have

$$\mathbb{E}[f(\mathbf{x}_{t+1})]$$

$$\le \mathbb{E}[f(\mathbf{x}_t)] - \gamma \mathbb{E}[g(\mathbf{x}_t)] + \gamma D \cdot \mathbb{E}\|\nabla f(\mathbf{x}_t) - \mathbf{q}_t\|_2 + \frac{LD^2\gamma^2}{2}$$

$$\le \mathbb{E}[f(\mathbf{x}_t)] - \gamma \mathbb{E}[g(\mathbf{x}_t)] + \gamma D \cdot \left(\|\nabla f(\mathbf{x}_t) - \mathbb{E}[\mathbf{q}_t]\|_2 + \mathbb{E}\|\mathbf{q}_t - \mathbb{E}[\mathbf{q}_t]\|_2\right) + \frac{LD^2\gamma^2}{2},$$

$$\le \mathbb{E}[f(\mathbf{x}_t)] - \gamma \mathbb{E}[g(\mathbf{x}_t)] + \gamma D \cdot \left(\|\nabla f(\mathbf{x}_t) - \mathbb{E}[\mathbf{q}_t]\|_2 + \sqrt{\mathbb{E}\|\mathbf{q}_t - \mathbb{E}[\mathbf{q}_t]\|_2^2}\right) + \frac{LD^2\gamma^2}{2}$$

$$\le \mathbb{E}[f(\mathbf{x}_t)] - \gamma \mathbb{E}[g(\mathbf{x}_t)] + \gamma D \cdot \left(\|\nabla f(\mathbf{x}_t) - \nabla f_\delta(\mathbf{x}_t)\|_2 + \sqrt{\frac{4d(C_g + LD)^2 + \delta_t^2 L^2 d^2}{2b}}\right)$$

$$\quad + \frac{LD^2\gamma^2}{2},$$

$$\le \mathbb{E}[f(\mathbf{x}_t)] - \gamma \mathbb{E}[g(\mathbf{x}_t)] + \gamma D \cdot \left(\frac{\delta_t Ld}{2} + \frac{2\sqrt{d}(C_g + LD) + \delta_t Ld}{\sqrt{2b}}\right) + \frac{LD^2\gamma^2}{2},$$

where the second inequality follows from triangle inequality, the third inequality is due to Jenson's inequality and the last inequality holds due to Lemma 4.6.

Summation over $t$ of the above inequality, we obtain

$$\mathbb{E}[f(\mathbf{x}_T)]$$
$$\leq f(\mathbf{x}_0) - \sum_{t=0}^{T-1} \gamma \mathbb{E}[g(\mathbf{x}_t)] + \gamma DT \left( \frac{\delta_t Ld}{2} + \frac{2\sqrt{d}(C_g + LD) + \delta_t Ld}{\sqrt{2b}} \right) + \frac{TLD^2\gamma^2}{2}$$
$$\leq f(\mathbf{x}_0) - \gamma T g_a + \gamma DT \left( \frac{\delta_t Ld}{2} + \frac{2\sqrt{d}(C_g + LD) + \delta_t Ld}{\sqrt{2b}} \right) + \frac{TLD^2\gamma^2}{2},$$

where the second inequality follows from the definition of $\widetilde{g}_a$. Note that by the zeroth-order optimality, we have $f(\mathbf{x}_{t+1}) \geq f(\mathbf{x}^*)$. Rearrange the above inequality we obtain

$$\mathbb{E}[g_a] \leq \frac{f(\mathbf{x}_0) - f(\mathbf{x}^*)}{T\gamma} + \frac{LD^2\gamma}{2} + D \left( \frac{\delta_t Ld}{2} + \frac{2\sqrt{d}(C_g + LD) + \delta_t Ld}{\sqrt{2b}} \right)$$
$$\leq \frac{D}{\sqrt{2T}} \left( \sqrt{L(f(\mathbf{x}_0) - f(\mathbf{x}^*))} + 2(L + C_g + LD) \right),$$

where the second inequality is achieved by setting $\gamma = \sqrt{2(f(\mathbf{x}_0) - f(\mathbf{x}^*))/(LD^2T)}$, $b = Td$ and $\delta_t = \sqrt{2/Td^2}$.

$\square$

## C  PARAMETERS SETTINGS FOR SECTION 5

For Frank-Wolfe white-box attack algorithm, we list the parameters we use in Section 5 at Table 5.

Table 5: Parameters used in Frank-Wolfe white-box attack.

| PARAMETER | $L_2$ CASE | $L_{\text{inf}}$ CASE |
|---|---|---|
| $T$ | 1000 | 1000 |
| $\{\gamma_t\}$ | 0.03 | 0.005 |
| $\lambda$ | 20 | 5 |

Similarly, for Frank-Wolfe black-box attack algorithm, we also list the parameters we use in Section 5 at Table 6.

Table 6: Parameters used in Frank-Wolfe black-box attack.

| PARAMETER | $L_2$ CASE | $L_{\text{inf}}$ CASE |
|---|---|---|
| $T$ | 10000 | 10000 |
| $\{\gamma_t\}$ | $0.05/\sqrt{t}$ | $0.03/\sqrt{t}$ |
| $\lambda$ | 50 | 30 |
| $b$ | 25 | 25 |
| $\delta_t$ | 0.001 | 0.01 |

We also list the hyperparameters we use for baseline algorithms. Specifically, for PGD, we set a step size of 0.05 for $L_2$ case and 0.01 for $L_\infty$ case. For CW, we set a step size of 0.002 for $L_2$ case step size of 0.005 for $L_\infty$ case. The confidence is set to 0 and we perform 10 times binary search for the constant starting from 0.01 ($L_2$ case) and 0.001 ($L_\infty$ case). For EAD, we use a step size of 0.01 and the same binary search strategy as CW and $\beta$ is set to be 0.001. In terms of black-box experiments, for ZOO, we set a step size of 0.01 and the initial constant is set to be 1 without binary search to achieve better query complexity. For NES-PGD, we set a step size of 0.3 for $L_2$ case and 0.01 for $L_\infty$ case.

Table 7: Comparison of $L_2$ norm based white-box attacks on ResNet V2 model with $\epsilon = 5$. We report attack success rate, average time and average distortion.

| METHODS | SUCCESS RATE (%) | AVERAGE TIME (s) | AVERAGE DISTORTION |
|---------|------------------|------------------|--------------------|
| PGD | 99.8 | 168.2 | **0.88** |
| CW | 98.6 | 278.8 | 1.45 |
| EAD | 73.0 | 109.2 | 2.81 |
| FW-White | 100.0 | **47.1** | 0.93 |

Table 8: Comparison of $L_\infty$ norm based white-box attacks on ResNet V2 model with $\epsilon = 0.05$. We report attack success rate, average time and average distortion.

| METHODS | SUCCESS RATE (%) | AVERAGE TIME (s) | AVERAGE DISTORTION |
|---------|------------------|------------------|--------------------|
| PGD | 100.0 | 26.8 | **0.0031** |
| CW | 100.0 | 538.9 | 0.0251 |
| FW-White | 100.0 | **14.9** | **0.0031** |

## D  ADDITIONAL EXPERIMENTS

### D.1  RESNET V2 WHITE-BOX ATTACK RESULTS

In this subsection, we present the white-box attack experiments on ResNet V2 model. Tables 7 and 8 present our experimental results for $L_2$ norm and $L_\infty$ norm based white-box attacks respectively. For the other baselines in the $L_2$ norm case, surprisingly, CW method cannot achieve the best $L_2$ distortion as it does in Inception V3 model. EAD method is relatively faster than CW in terms of attack time yet it has the largest distortion and a quite low success rate of $73.0\%$. PGD has the smallest average distortion in this setting, yet it also costs a lot of attack time. On the other hand, our proposed algorithm achieves the highest attack success rate within very short attack time with very small distortion. It significantly reduces the time complexity needed for effective attacking data with large dimensionality. For the $L_\infty$ norm case, CW method takes significantly longer time and does not perform very well on average distortion either. Our proposed white-box attack algorithm, on the other hand, again achieves the shortest average attack time and 100% success rate.

### D.2  RESNET V2 BLACK-BOX ATTACK RESULTS

Table 9: Comparison of $L_2$ norm based black-box attacks on ResNet V2 model with $\epsilon = 5$. Query limit is set to be $50,000$. We report attack success rate, average time and average number of queries needed per image. Opt I and Opt II refer to the two options in Algorithm 2.

| METHODS | SUCCESS RATE (%) | AVERAGE TIME (s) | AVERAGE QUERIES |
|---------|------------------|------------------|-----------------|
| ZOO | 2.4 | 696.0 | 49495.2 |
| NES-PGD | 58.0 | 75.3 | 36748.1 |
| FW-Black (Opt I) | 57.4 | 75.0 | 34382.5 |
| FW-Black (Opt II) | **58.8** | **74.7** | **34362.8** |

Table 10: Comparison of $L_\infty$ norm based black-box attacks on Inception V3 model with $\epsilon = 0.05$. Query limit is set to be $50,000$. We report attack success rate, average time and average number of queries needed per image. Opt I and Opt II refer to the two options in Algorithm 2.

| METHODS | SUCCESS RATE (%) | AVERAGE TIME (s) | AVERAGE QUERIES |
|---------|------------------|------------------|-----------------|
| NES-PGD | 90.4 | 44.7 | 20914.0 |
| FW-Black (Opt I) | 90.8 | 44.1 | **19934.6** |
| FW-Black (Opt II) | **91.6** | **44.0** | 20004.6 |

In this subsection, we present the black-box experiments on ResNet V2 model. We again mainly focus on evaluating attack success rate, time and number of queries needed. In previous experiments on Inception V3 model, we show the performance of different black-box attack algorithms given enough number of queries (i.e., 500,000 per attack per image). And it shows that basically all

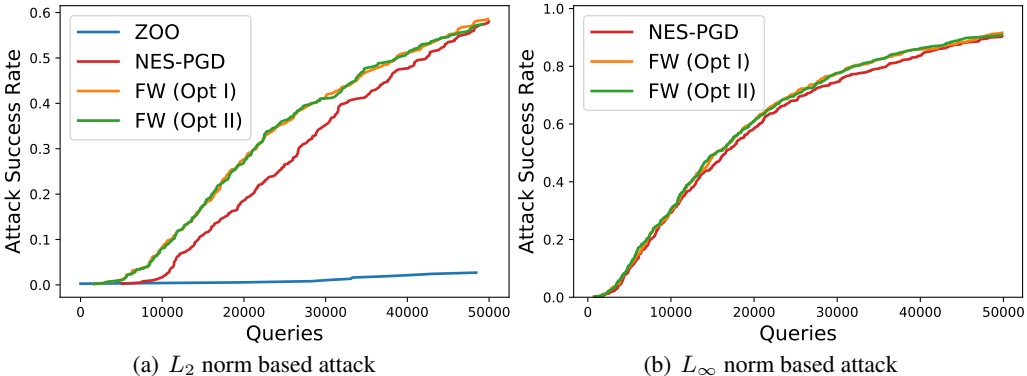

(a) $L_2$ norm based attack        (b) $L_\infty$ norm based attack

Figure 3: Attack success rate against the number of queries plot for different algorithms in both $L_2$ norm and $L_\infty$ norm based black-box attacks on ResNet V2 model.

algorithms can achieve very high attack success rate (almost 100%). Now we examine a much harder case, where we reduce the the number of allowed queries per attack per image to only $50,000$. Tables 9 and 10 present our experimental results for $L_2$ norm and $L_\infty$ norm based black-box attacks respectively. We still set $\epsilon = 5$ for $L_2$ case and $\epsilon = 0.05$ for $L_\infty$ case.

For the $L_2$ norm case, ZOO method barely succeeds due to the strict query limit of $50,00$ while it typically requires over $10^6$ queries to attack successfully. Our proposed Frank-Wolfe black-box attacks, on the other hand, achieve nearly $60\%$ attack success rate under such a stringent query budget. For the $L_\infty$ norm case, both NES-PGD method and ours achieve over $90\%$ success rate. Even though they share similar average attack time and average number of queries needed, our Frank-Wolfe based methods still achieve the best in terms of all three evaluation metrics.

Figure 3 illustrates the attack success rate against the number of queries plot for different algorithms in both $L_2$ norm and $L_\infty$ norm based black-box attack on ResNet V2 model. Note that here we have a query limit of $50,000$, which is especially hard for the $L_2$ norm case. As we can see from the Figure 3, our proposed Frank-Wolfe black-box attack algorithm (both options) achieves the best performance (highest attack success rate and smallest queries needed for achieving the same success rate).

### D.3 VISUALIZATION EXAMPLES

For the completeness, we also provide some visual illustrations on the adversarial examples generated by various algorithms. Figure 4 shows some adversarial examples generated through different $L_2$ norm based white-box attacks. Figure 5 shows some adversarial examples generated through different $L_\infty$ norm based black-box attacks.

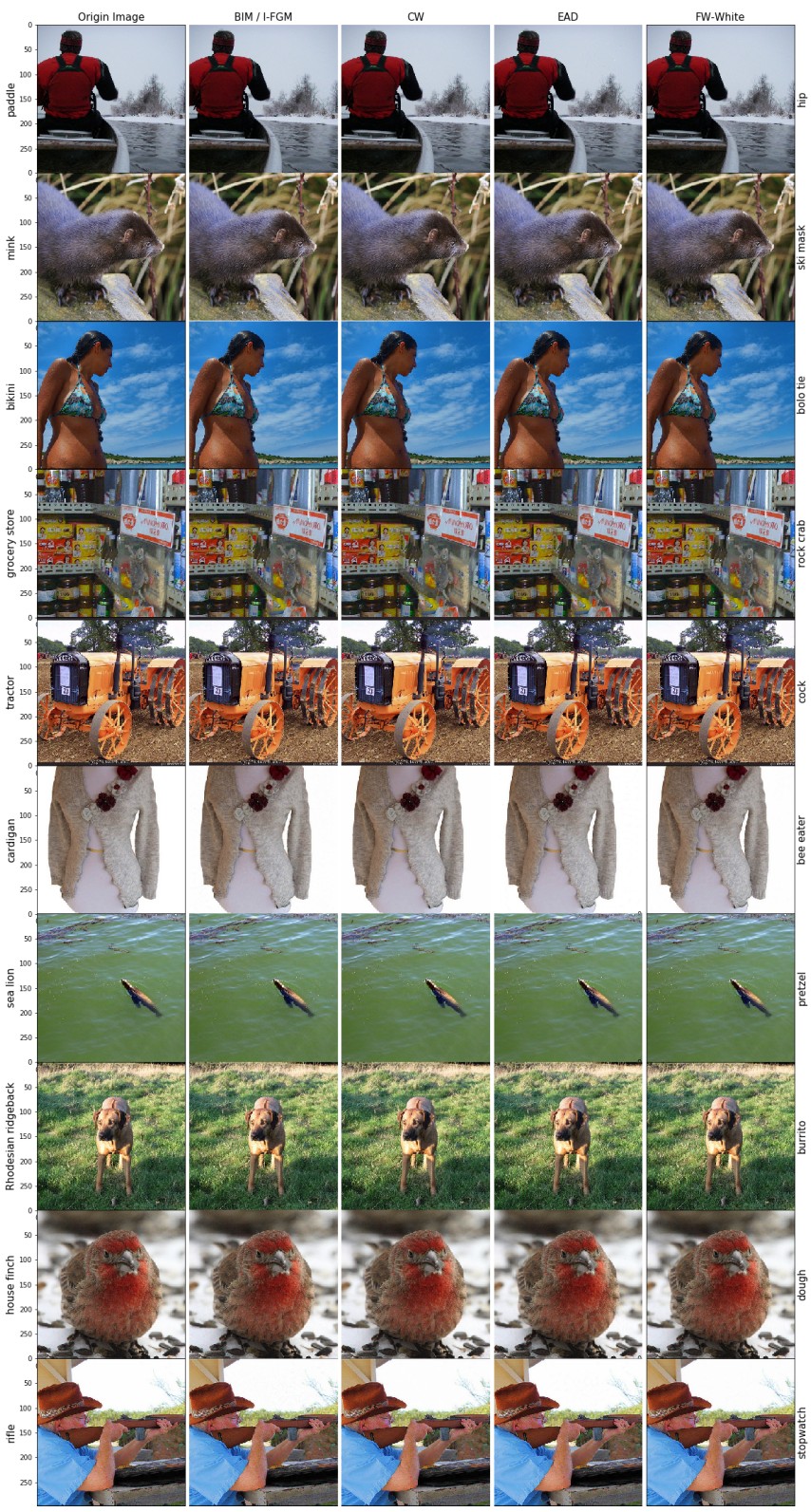

Figure 4: Sample adversarial examples generated through different $L_2$ norm based white-box attacks. Left side labels denote the original class label and right side labels denote the target class label.

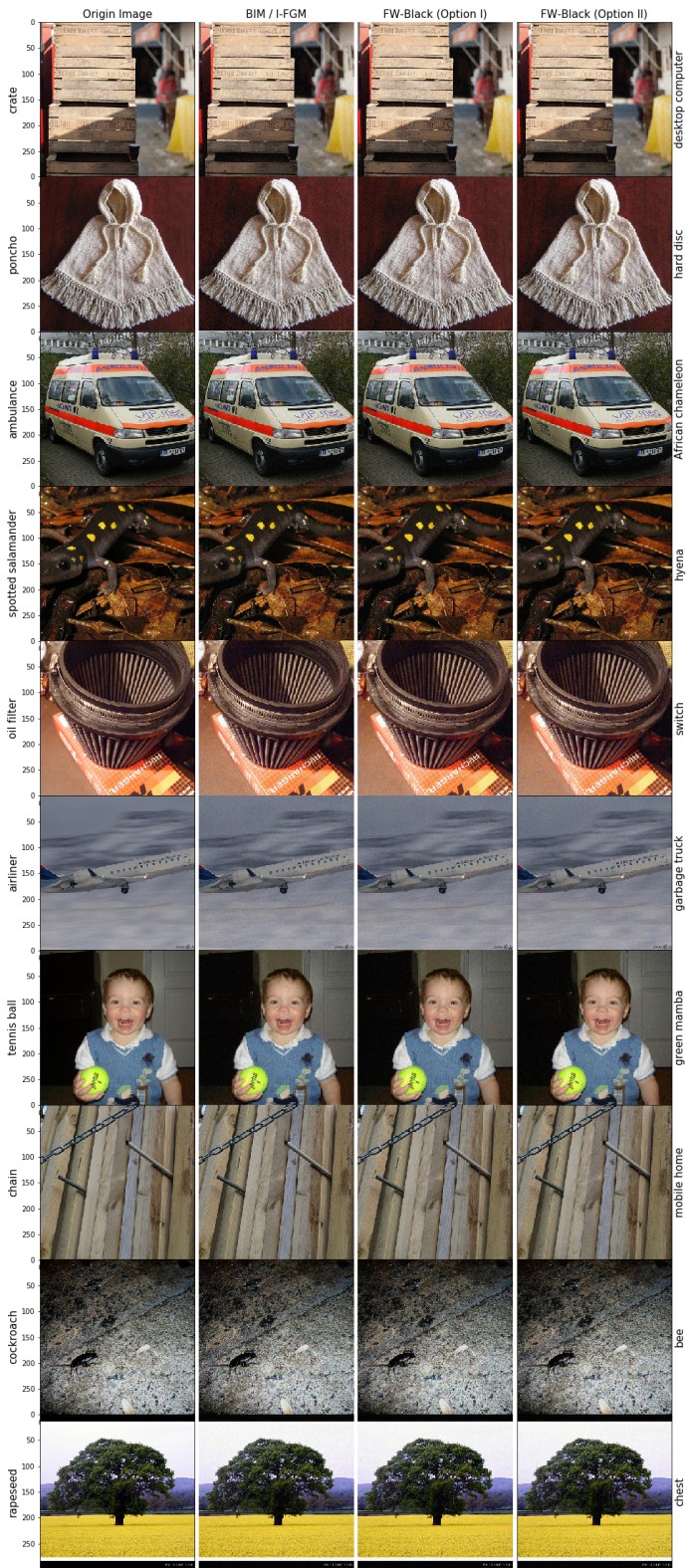

Figure 5: Sample adversarial examples generated through different $L_\infty$ norm based black-box attacks. Left side labels denote the original class label and right side labels denote the target class label.

