# OpenReview forum: "A Frank-Wolfe Framework for Efficient and Effective Adversarial Attacks"
_ICLR.cc/2019/Conference_

### Official Review · AnonReviewer1 · 2018-10-31
**Interesting paper, a bit problematic experimental set-up**

**Rating:** 7
**Confidence:** 4

**Review:**

The paper proposes using the Frank-Wolfe algorithm for fast adversarial attacks. They prove upper bounds on the Frank-Wolfe gap and show experimentally that they can attack successfully much faster than other algorithms. In general I find the paper novel (to the best of my somewhat limited knowledge), interesting and well written. However I find the white-box experiments lacking as almost every method has 100% success rate. Fixing this would significantly improve the paper.

Main remarks:
- Need more motivation for faster white-box attack. One good motivation for example is adversarial training, e.g. Kurakin et al 2017 ‘ADVERSARIAL MACHINE LEARNING AT SCALE’ that would benefit greatly from faster attacks

- White-box attack experiments don’t really prove the strength of the method, even with imagenet experiments, as almost all attacks get 100% success rate making it hard to compare. Need to compare in more challenging settings where the success rate is meaningful, e.g. smaller epsilon or a more robust NN using some defence. Also stating the 100% success rate in the abstract is a bit misleading for the this reason.

-Something is a bit weird with the FGM results. While it is a weaker attack, a 0%/100% disparity between it and every other
attack seems odd.

-The average distortion metric (that’s unfavourable to your method anyway) doesn’t really mean anything as the constraint optimization has no incentive to find a value smaller than the constraint.

- Regarding lambda>1, you write that “we argue this modification makes our algorithm more general, and gives rise to better attack results”. I did not see any theoretical or empirical support for this in the paper. Also, it seems quite strange to me that making the FW overshot and then projecting back would be beneficial. Some intuitive explanation on why this should help and/or empirical comparison would be a great addition.

- The authors claim that this is the first zeroth-order non-convex FW convergence rate, I am not familiar enough with the field to evaluate this claim and its significance.

- Alg. 1 for T>1 is very similar to I-FGM, but also ‘pulls’ x_t towards x_orig. It would be very useful to write the update more explicitly and compare and contrast this 2 very similar updates. This gives nice insight into why this should intuitively work better.

- I am not sure what the authors mean by “the Frank-Wolfe gap is affine invariant”. If we scale the input space by a, the gap should be scaled by a^2 - how/why is it invariant?

- I am not sure what you mean in 5.4 “we omit all grid search/ binary search steps…”

Minor remarks:
- In remark 4.8 in the end option I and II are inverted by mistake

- In 5.1, imagenet results are normally top-5 error rate not top-1 acc, would be better to report that more familiar number.

- In the proof you wrongfully use the term telescope sum twice, there is nothing telescopic about the sum it is just bound by the max value times the length.

---

> ### Author Response · Authors · 2018-11-26
> **Response to Reviewer1**
>
> 1. Thank you for your suggestion, we have addressed this in the revision as you suggested. This is indeed a good motivation.
>
> 2. Thank you for your suggestion. We have further added experiments using even stronger query limit (previously 500000, now 50000) for the additional experiments on ResNet V2 model in the supplemental material. (We did not choose to use smaller epsilon because first, we already used a quite standard choice of epsilon, second, as you said, going for extremely small distortion does not really mean anything in adversarial context.) As you can see, in this even harder setting our proposed algorithm still maintain a performance lead over other baselines. Also, we have revised the statement in the abstract as you suggested.
>
> 3. You are right, it is a quite weak attack and we have removed it from the table (just mention it in the text).
>
> 4. Yes, we could just remove the distortion column in our result. We choose to include it because we do not want others to think that we actually trade a lot of distortions (to make problem easy) for speedup in runtime.
>
> 5. We have added further empirical evidence to show that in the revision. From an intuitive perspective, using lambda>1 is essentially a “relax and tighten” step by first relax the constraint to make the problem easier, and then tighten it back to the real constraint. The “relax and tighten” idea has been widely used in constrained optimization, and we adapted this idea to Frank-Wolfe algorithm to make it even faster.
>
> 6. As mentioned in an anonymous comment, there is one paper which proposed a similar but different zeroth-order non-convex FW algorithm as well as convergence rate analysis ahead of us.  We were not aware of this paper when we prepared our ICLR submission, since it was posted only ten days before the ICLR deadline. We have cited this paper and modify our claim correspondingly in the revision. Nevertheless, it does not affect the main contribution of our paper: a novel Frank-Wolfe based adversarial attack framework for both white-box and black-box attacks, which is much more efficient than existing white-box/black-box adversarial attacks in both query complexity and runtime.
>
> 7. Thank you for your suggestion and we have explicitly written down the update for a better comparison in the supplemental materials (Section A) in the revision.
>
> 8. It means it is invariant to an affine transformation of the constraint set, i.e., if we choose to re-parameterize of the constraint with some linear or affine transformation M, the original and the new optimization problem will looks the same to the Frank-Wolfe algorithm. Please refer to [Jaggi (2013)], [Lacoste-Julien (2016)] for more details.
>
> 9. In white-box setting, we perform grid search / binary search for parameter epsilon (or c for CW) for all algorithms. This will lead to better/ closer distortions for all methods. In black-box setting, we care more about query complexity and thus did not perform the grid search/binary search steps to avoid extra queries in finding the best epsilon/lambda.
>
> 10. Thank you for pointing these typos out, we have addressed it in the revision.

---

> > ### Comment · AnonReviewer1 · 2018-11-27
> > **Thank you for the revision**
> >
> > With the updates and mostly with new experiments where not all attacks reach 100% accuracy, I think this paper should be accepted.

---

> > > ### Author Response · Authors · 2018-11-27
> > > **Thank you**
> > >
> > > Thank you very much for reading our response and revision, and increasing your score.

---

### Official Review · AnonReviewer2 · 2018-11-02
**A method to produce adversarial attack using a Frank-Wolfe inspired method**

**Rating:** 5
**Confidence:** 4

**Review:**

This paper provide a method to produce adversarial attack using a Frank-Wolfe inspired method.

I have some concerns about the motivation of this method:
 - What are the motivations to use Frank-Wolfe ? Usually this algorithm is used when the constraints are to complicated to have a tractable projection (which is not the case for the L_2 and L_\infty balls) or when one wants to have sparse iterates which do not seem to be the case here.
 - Consequently why did not you compare simple projected gradient method ? (BIM) is not equivalent to the projected gradient method since the direction chosen is the sign of the gradient and not the gradient itself (the first iteration is actually equivalent because we start at the center of the box but after both methods are no longer equivalent).
 - There is no motivations for the use of $\lambda >1$ neither practical or theoretical since the results are only proven for $\lambda =1$ whereas the experiments are done with \lambda = 5,20 or 30.
 - What is the difference between the result of Theorem 4.3 and the result from (Lacoste-Julien 2016)?

Depending on the answer to these questions I'm planning to move up or down my grade.

 In the experiment there is no details on how you set the hyperparameters of CW and EAD. They use a penalized formulation instead of a constrained one. Consequently the regularization hyperparameters have to be set differently.

 The only new result seem to be Theorem 4.7 which is a natural extension to theorem 4.3 to zeroth-order methods.

Comment:
- in the whole paper there is $y$ which is not defined. I guess it is the $y_{tar}$ fixed in the problem formulation Sec 3.2.  In don't see why there is a need to work on any $y$. If it is true,  case assumption 4.5 do not make any sense since $y = y_{tar}$ (we just need to note $\|\nabla f(O,y_{tar})\| = C_g$) and some notation could be simplified setting for instance $f(x,y_{tar})  = f(x)$.
- In Theorem 4.7 an expectation on g(x_a) is missing

Minor comments:
- Sec 3.1 theta_i -> x_i
- Sec 3.3 the argmin is a set, then it is LMO $\in$ argmin.

===== After rebuttal ======
The authors answered some of my questions but I still think it is a borderline submission.

---

> ### Author Response · Authors · 2018-11-26
> **Response to Reviewer2**
>
> Thank you for your helpful comments!
>
> 1. You are right that Frank-Wolfe would be advantageous over PGD when the constraints are more complicated and adversarial attack may not be such a case. Yet it is also well-known that Frank-Wolfe has quite different optimization behavior compared with PGD even though they have the same order of convergence rate. Therefore, it is interesting and important to examine the performance of Frank-Wolfe algorithm for adversarial attack, given the fact that PGD has been shown to be a very effective for adversarial attack. In fact, from our work, we found that Frank-Wolfe based methods are generally more efficient than PGD method.
> From another perspective, Frank-Wolfe solves the problem by calling Linear Minimization Oracle (LMO) over the constraint set at each iteration. This LMO shares the same intuition as FGSM, which also tries to linearize the neural network loss function to find the adversarial examples. In this sense, it is a quite natural attempt to revisit FGSM under the Frank-Wolfe framework.
>
> 2. We are sorry maybe we didn’t explain it very well in the paper, but this is a misunderstanding. We indeed compared our method with generalized I-FGSM/BIM, which is exactly the same as PGD (In [Madry et al.] they also mentioned this in Section 2.1 and they refer it as FGSM^k).  We decide to just call it PGD in the revision to avoid confusion. We hope this remove your concern.
>
> 3. Indeed, theoretically we can only prove for $\lambda$ = 1 case. Yet we found that larger \lambda brings us more speedup.  We have added further empirical evidence (performance comparison of our method with different \lambda in Figure 1 in the revised paper) to justify it. Intuitively speaking, using lambda>1 is essentially a “relax and tighten” step by first relax the constraint to make the problem easier, and then tighten it back to the real constraint. The “relax and tighten” idea has been widely used in constrained optimization, and in this paper we adapted this idea into Frank-Wolfe algorithm to make it even faster.
>
> 4. [Lacoste-Julien 2016] considered the general first-order Frank-Wolfe algorithm for nonconvex smooth optimization. The result of Theorem 4.3 in our paper is almost the same as the result in (Lacoste-Julien 2016), except that the choices the learning rate in these two papers are different though. We have made it clear in the revision.
>
> 5. We have added detailed hyperparameter settings for CW and EAD in the revision in the supplemental materials.
>
> 6. While Theorem 4.7 is new and may be of independent interest in the optimization community,  it is not the main contribution in this paper. We would like to emphasize that our major contribution in this paper is a Frank-Wolfe based algorithm for adversarial attack, which is more efficient than PGD based adversarial attack algorithm and other baselines.
>
> 7. Sorry about the confusion. $y$ should be replace by $y_{tar}$. It is a simplified notation we mentioned in the proof in the appendix. Thank you for your suggestion and we have revised the notation $f(x,y_{tar})$ to $f(x)$.
>
> 8. Thank you for pointing out several typos. We have fixed all of them in the revision.

---

> > ### Comment · AnonReviewer2 · 2018-11-29
> > **Comments on authors' response**
> >
> > 2. FGSM^k is **not** projected gradient descent on the objective you are optimizing because the signal considered is the **sign** of the gradient instead of the gradient itself.
> >
> > Considering the sign of the gradient provides a method that may diverge for any step-size $\eta$:
> > Take for instance $\L(\theta) = \theta^2, \theta_0= \eta/2$ and the constraint to clip in $[-1,1]$ then since $sign(\nabla \L (\theta) = sign(\theta)$ we have that $\theta_1 = \theta_0 -  \eta \sign(\nabla \L(\theta_0) = \eta/2 - \eta = - \eta/2$ and then $\theta_2 = \eta/2$ providing a sequence that oscillate between $\eta/2$ and $-\eta/2$.
> >
> > 4. The step-size you are proposing might be challenging to compute (since you may not have access to f(x^*) ) whereas the step-size proposed by Lacoste-julien (Option II) depends on the same constants as yours an the gap function with is very cheap to compute (since you have  already computed $d_t$)

---

> > > ### Author Response · Authors · 2018-11-30
> > > **Response to Reviewer2's new comments**
> > >
> > > Thank you for reading our response and revision.
> > >
> > > 2. We apologize we misunderstood your original comment on PGD. Thank you for your clarification. Now we know where the misunderstanding comes from: we are actually talking about different “PGD” methods. The fact you describe is indeed correct, and you are referring to the standard PGD method in constrained optimization, let us call it “standard PGD”. However, what we refer to, is the “adversarial PGD” proposed by [Madry et al.]. The only difference is that in “adversarial PGD”, the gradient should be first “normalized” before one can use it. In $L_\infty$ norm case, the “normalized” gradient is the sign of the gradient and in $L_2$ norm case, the “normalized” gradient is the gradient normalized by its $L_2$ norm. In this way,  “adversarial PGD” is the same as FGSM^k in $L_\infty$ norm case as pointed out in [Madry et al.].
> > >
> > > We will add  “standard PGD” as a baseline in our white-box attack experiments to address your question. Running the experiments could take a while and we will try to get the new results posted here within the next 48 hours. We hope this could clear your concern.
> > >
> > > 4. You are right we do not have access to f(x^*).  However, in [Lacoste-julien] (Option II), their step size also depends on C > C_f, where C_f, by definition, is also intractable to compute. In fact, both our theorem and [Lacoste-julien]’s theorem rely on the smoothness assumption which is also not satisfied in real deep learning tasks due to the use of RELU activations and max-pooling layers. Therefore, in our experiments, we did not use the “theoretical” step size as suggested by the theorem.  Actually, we use $\sqrt(c/(T))$ as the step size and performed grid search to tune c.   We found that constant step size is more than enough to give us good performance in white-box attacks.
> > >
> > > In the end, we are not arguing whose step size is better, rather, we just want to provide a theorem describing the convergence behavior of Frank-Wolfe white-box attack algorithm, for completeness.

---

> > > > ### Author Response · Authors · 2018-12-05
> > > > **Follow up with Reviewer 2**
> > > >
> > > > Dear Reviewer 2, thank you for reading our response and increasing your score. At the end of your updated review, you said that “the authors answered some of my questions”. We apologize if we missed any of your questions. We wonder could you let us know what are the questions that we did not answer? We will answer them asap. Thank you!
> > > >
> > > > Best,
> > > > Authors

---

> > > ### Author Response · Authors · 2018-12-02
> > > **[UPDATE] The experimental results of standard PGD baseline**
> > >
> > > [UPDATE] We have performed the additional baseline “standard PGD” for white-box attack. All test show that “standard PGD” can also achieve 100% attack success rate in white-box attack, for both $L_2$ norm and $L_\infty$ norm cases, and for both Inception and ResNet networks.
> > >
> > > The new tables containing “standard PGD” results can be found in the following anonymous link: https://www.dropbox.com/sh/p2uj97yixobnku2/AAC0o9dJDtt-cvQg7pcjQXila?dl=0
> > >
> > > It can be seen that “standard PGD” has similar performance with “adversarial PGD” in our experiments. It does not change any of our conclusions. We will update these tables in the final version of our paper (We are not allowed to revise our submission at this point). Please let us know if you have any other suggestion.

---

### Official Review · AnonReviewer3 · 2018-11-06
**Promising results but some questions about experiments**

**Rating:** 5
**Confidence:** 4

**Review:**

The paper investigates the Frank-Wolfe (FW) algorithm for constructing adversarial examples both in a white-box and black-box setting. The authors provide both a theoretical analysis (convergence to a stationary point) and experiments for an InceptionV3 network on ImageNet. The main claim is that the proposed algorithm can construct adversarial examples faster than various baselines (PGD, I-FGSM, CW, etc.), and from fewer queries in a black-box setting.

The FW algorithm is a classical method in optimization, but (to the best of my knowledge) has not yet been evaluated yet for constructing adversarial examples. Hence it is a natural question to understand whether FW performs significantly better than current algorithms in this context. Indeed, the authors find that FW is 6x - 20x faster for constructing white-box adversarial examples than a range of relevant baseline, which is a significant speed-up. However, there are several points about the experiments that are unclear to me:

- It is well known that the running times of optimization algorithms are highly dependent on various hyperparameters such as the step size. But the authors do not seem to describe how they chose the hyperparameters for the baselines algorithms. Hence it is unclear how large the running time improvement is compared to a well-tuned baseline.

- Other algorithms in the comparison achieve a better distortion (smaller perturbation). Since finding an adversarial with smaller perturbation is a harder problem, it is unclear how the algorithms compare for finding adversarial examples with similar distortion. Instead of reporting a single time-vs-distortion data point, the authors could show the full trade-off curve.

- The authors only provide running times, not the number of iterations. In principle all the algorithms should have a similar bottleneck in each iteration (computing a gradient for the input image), but it would be good to verify this with an iteration count vs success rate (or distortion) plot. This would also allow the authors to compare their theoretical iteration bound with experimental data.

In addition to these three main points, the authors could strengthen their results by providing experiments on another dataset (e.g., CIFAR-10) or model architecture (e.g., a ResNet), and by averaging over a larger number of test data points (currently 200).

Overall, I find the paper a promising contribution. But until the authors provide a more thorough experimental evaluation, I hesitate to recommend acceptance.


Additional comments:

The introduction contains a few statements that may paint an incomplete or confusing picture of the current literature in adversarial attacks on neural networks:

* The abstract claims that the poor time complexity of adversarial attacks limits their practical usefulness. However, the running time of attacks is typically measured in seconds and should not be the limiting element in real-world attacks on deep learning systems. I am not aware of a setting where the running time of an attack is the main computational bottleneck (outside adversarial training).

* The introduction distinguishes between "gradient-based methods" and "optimization-based methods". This distinction is potentially confusing to a reader since the gradient-based methods can be seen as optimization algorithms, and the optimization-based methods rely on gradients.

* The introduction claims that black-box attacks need to estimate gradients coordinate-wise. However, this is not the case already in some of the prior work that uses random directions for estimating gradients (e.g., the cited paper by Ilyas et al.)

I encourage the authors to clarify these points in an updated version of their paper.

---

> ### Author Response · Authors · 2018-11-26
> **Response to Reviewer3**
>
> Thank you for your constructive comments!
>
> 1. We fully understand your concern and we have added detailed description in the supplemental materials to show the hyperparameters we use for baseline methods in the revision.
>
> 2. We would like to argue that constrained optimization based formulation itself is not designed to achieve better distortion compared with regularized optimization based formulation. So there is no surprise that our algorithm’s distortion is not the best. On the other hand, as mentioned by the other reviewer, distortion is usually not that essential in adversarial attacks as long as it is maintained in a reasonable range. We could actually remove the distortion column, instead, we chose to include it just to show that we did not trade a lot of distortions (to make problem much easier) and thus gains speedup. From our experimental results, you can see that our proposed method achieves significant speedup while keeping the distortion around the same level as the best baselines.
>
> 3. Thank you for your suggestion. We have further added success rate vs queries plot (for black-box case) and loss vs iterations plot (for white-box case) in the revision. As you can see, in terms of number of iterations / queries, our method still outperforms the other baselines by a large margin.
>
> 4. Thank you for your suggestion. We have further added experiments on ResNet V2 model and averaging over 500 correctly classified pictures to strengthen our result. Again, this additional experiments show that our method outperforms the other baselines for both white-box attack and black-box attack.
>
> 5. Regarding poor time complexity in practice, first, as you mentioned, adversarial training currently is quite slow due to the slow adversarial attack steps. Better time complexity of adversarial attack could significantly speed up adversarial training algorithms. Second, it is worth noting that the running time complexity of adversarial attack also highly depends on the input size. For example, if you attack a CIFAR-10 classifier or an MNIST classifier, it could take only seconds per attack even for the slowest algorithm since the input size is only 32 by 32 (or 28 by 28). However, if you attack a ImageNet classifier or even higher dimensional data classifier, it could take significantly longer time (minutes). That is why reducing the runtime of adversarial attack is very important.
>
> 6. We apologize for this confusion. Regarding “gradient-based” / “optimization based” methods and coordinate-wise black-box attacks, we have changed our description to avoid confusion. Thank you for pointing it out.

---

> > ### Comment · AnonReviewer3 · 2018-12-01
> > **How was the PGD step size selected?**
> >
> > I would like to thank the authors for their detailed reply.
> >
> > Did you also tune the step size of PGD for the experiments in Figure 1 for instance? To make a fair comparison, both FW and PGD should be tuned appropriately. In particular, the authors should ensure that the chosen step size is not at the boundary of the explored range of hyperparameters.

---

> > > ### Author Response · Authors · 2018-12-02
> > > **Response to Reviewer3's new comments**
> > >
> > > Thank you for reading our response and revision.
> > >
> > > Yes, we performed grid search to tune the step size for all baselines algorithm to ensure a fair comparison. We are sure the chosen step size is not at the boundary. In detail, we first tune the step size by searching the grid {0.0001, 0.001, 0.01, 0.1, 1,10}, and find the best step size over this grid. It turns out that the boundary points 0.0001 and 10 were never selected to be the best step size.  Given the selected best step size in the previous grid (e.g., 0.1), we evenly splitting the intervals before and after this step size (e.g., 0.01-0.1, and 0.1-1) into 10 subintervals respectively to construct a refined grid (e.g., {0.01,0.02,..., 0.09, 0.1, 0.2, …,0.9,1}), and fine-tune the step size by search this refined grid. We will elaborate it in the final version of our paper.

---

> > > ### Author Response · Authors · 2018-12-05
> > > **Follow up with Reviewer 3**
> > >
> > > Dear Reviewer 3, we have posted our latest response on OpenReview to address your question. We’d like to follow up with you to see whether you had a chance to look it over? We hope our responses could clear your concerns.  Thank you!
> > >
> > > Best,
> > > Authors

---

### Public Comment · (anonymous) · 2018-10-03
**related paper**

https://arxiv.org/abs/1809.06474 - this paper in nips already provides rates for zeroth-order non-convex Frank-Wolfe type algorithm.

---

> ### Author Response · Authors · 2018-10-12
> **Reply to “related paper on zeroth-order non-convex Frank-Wolfe type algorithm”**
>
> We were not aware of this paper when we prepared our ICLR submission, since it was posted only ten days before the ICLR deadline. We will cite this paper and modify our claim correspondingly in a later version. Nevertheless, it does not affect the main contribution of our paper: a novel Frank-Wolfe based adversarial attack framework for both white-box and black-box attacks, which is much more efficient than the existing white-box/black-box adversarial attacks in both query complexity and runtime.
>
> In addition, we also would like to emphasize that the zeroth-order Frank-Wolfe algorithm we proposed is different from the algorithm proposed in the paper you pointed out. More specific,  they use one-side finite difference zeroth-order gradient estimator with standard Gaussian sensing vectors, while we use the two-side symmetric finite difference zeroth-order gradient estimator with sensing vectors sampled from the unit sphere.

---

### Meta-Review · Area_Chair1 · 2018-12-17
**Not ready for publication at ICLR**

**Confidence:** 4
**Recommendation:** Reject

**Metareview:**

While there was some support for the ideas presented, the majority of the reviewers did not think the submission is ready for publication at ICLR. Significant concerns were raised about clarity of the exposition.